# Examining the Effect of Radiant Exposure on Commercial Photopolimerizable Dental Resin Composites

**DOI:** 10.3390/dj6040055

**Published:** 2018-10-07

**Authors:** Carlos Enrique Cuevas-Suárez, Belinda Pimentel-García, Alejandro Rivera-Gonzaga, Carlos Álvarez-Gayosso, Adriana Leticia Ancona-Meza, Guillermo Grazioli, Eliezer Zamarripa-Calderón

**Affiliations:** 1Academic Area of Dentistry, Health Sciences Institute, Autonomous University of the Hidalgo State, Circuito Ex Hacienda La Concepción Km. 1.5. San Agustín Tlaxiaca, Hidalgo, C.P. 42160, Mexico; cecuevas@uaeh.edu.mx (C.E.C.-S.); piq03594@uaeh.edu.mx (B.P.-G.); jose_rivera10098@uaeh.edu.mx (A.R.-G.); ancona@uaeh.edu.mx (A.L.A.-M.); 2Faculty of Dentistry, National Autonomous University of México, Circuito de la Investigación Científica, Ciudad Universitaria, Delegación Coyoacán, Ciudad de México, C.P. 04510, Mexico; calvarezgayosso@comunidad.unam.mx; 3Department of Dental Materials, Faculty of Dentistry, University of the Republic, General Las Heras 1925, Montevideo, C.P. 11600, Uruguay; ggrazioli@odon.edu.uy

**Keywords:** composite resin, dental restoration, photopolymerization

## Abstract

The objective of this study was to evaluate the effect of radiant exposure on the chemical and physical properties of four commercial dental resin composites. The four dental resin composites used were Kalore, Admira, Tetric N-Ceram Bulk Fill, and Filtek Z350 XT. The composites were subjected to three curing protocols: 1000 mW/cm^2^ for 5 s, 1000 mW/cm^2^ for 10 s, and 400 mW/cm^2^ for 25 s. The flexural strength, elastic modulus, water sorption, water solubility, degree of conversion, and polymerization shrinkage were evaluated. The results were analyzed by means of ANOVA and Tukey tests. For Admira and Kalore, significant differences between light exposure protocols were observed for the elastic modulus (p < 0.001), which was higher when a higher amount of radiant exposure was used. For Filtek Z350, differences were only observed for the degree of conversion (p < 0.001), and a higher amount of radiant exposure allowed us to obtain higher values. The Tetric N-Ceram Bulk Fill analysis showed significant differences for the elastic modulus and water sorption (p < 0.001), and specimens that had been subject to a radiant exposure of 10 J/cm^2^ showed a higher elastic modulus. In most cases, the physical and mechanical properties analyzed were not affected by different radiant exposure values. Other resin-based composite (RBC) characteristics seem to have a greater influence on material properties.

## 1. Introduction

Due to their versatility and aesthetic advantages, photopolymerizable resin-based dental composites (RBCs) have been widely used in the restoration of dental caries or other tooth defects [1]. In order to achieve optimal material properties throughout the entire restoration, RBCs are placed using layering techniques [2]. The multi-increment technique is time-consuming, and there exists significant demand for a reduction in cure time to minimize the time it takes to complete the chairside procedure [3].

The photopolymerization efficiency of RBCs is influenced by several factors [4]. Among them, factors related to the photopolymerization unit have been widely studied; for example, with regard to light irradiance, the use of at least 400 mW/cm^2^ has been recommended [5,6]. Another directly related factor is radiant exposure, which is the product of irradiance (mW/cm^2^) and exposure time (s). Radiant exposure (J/cm^2^) has a direct relationship with the degree of conversion and other important mechanical properties of photopolymerizable RBCs [7,8]. The amount of radiant exposure required to adequately polymerize an increment of an RBC varies greatly, and there is still much confusion over which specific irradiance and time combination will provide optimal polymerization within a short clinical timeframe [3].

Based on the principle known as the “exposure reciprocity law”, which describes a reciprocal relationship between irradiance and exposure time to achieve equivalent polymerization of RBCs [9], there exists a tendency among dentists and manufacturers to use or suggest a high-power illumination to reduce the curing time [4]. It has been demonstrated that many resins appear to follow exposure reciprocity with regard to the degree of conversion, elastic modulus, hardness, and cure depth [10,11,12]. However, the validity of this rule has been challenged by other studies [13,14]. The available evidence for the exposure reciprocity law is contradictory because research groups have studied different resins, ranges of irradiance, and radiant exposures to investigate this phenomenon.

Recently, new types of photopolymerizable RBCs have been introduced, including low stress and low shrinkage composites, ormocer-based composites, and bulk-fill composites. These types of materials include new monomer, filler, or photoinitiator technologies that have not been fully studied. The introduction of new or modified dental products into the market requires a guarantee that these materials have at least similar properties to other RBCs.

Little is known about the effect of radiant exposure on these new types of materials. Consequently, the aim of this work was to evaluate the effect of radiant exposure on the physical and chemical properties of four resin-based composites. The null hypothesis tested was that similar properties will be achieved when composites are exposed to the same amount of radiant exposure (5 and 10 J/cm^2^) at different irradiance and time levels.

## 2. Materials and Methods

In this study, four dental resin composites were used: a low-shrink, low-stress composite (Kalore^TM^, GC Corporation, Tokyo, Japan), an ormocer-based composite (Admira^®^, Voco, Cuxhaven, Germany), a bulk-fill composite (Tetric^®^ N-Ceram Bulk Fill, Ivoclar-Vivadent, Schaan, Liechtenstein), and a nanofilled composite (Filtek^TM^ Z350 XT, 3M ESPE, St. Paul, MN, USA). Information on their manufacture is listed in Table 1.

### 2.1. Curing Protocols and Radiant Exposures

Composites were cured with a commercial light-emitting diode (LED) photopolymerization unit (LED Bluephase 16i, Ivoclar Vivadent, Schaan, Liechtenstein). They were subjected to three radiant exposure levels: Group A (1000 mW/cm^2^ for 5 s (5 J/cm^2^)); Group B (1000 mW/cm^2^ for 10 s (10 J/cm^2^)); and Group C (400 mW/cm^2^ for 25 s (10 J/cm^2^)). The intensity of light irradiation was monitored using a digital radiometer (Bluephase meter, Ivoclar, Vivadent).

### 2.2. Degree of Conversion

The degree of conversion of the composite materials was determined by infrared spectroscopy (Frontier, Perkin Elmer, Waltham, MA, USA). An infrared spectrum of the uncured and the cured samples was obtained. The measurement was made in real time using an attenuated total reflection (ATR) unit. For each spectrum, the height of the aliphatic C=C peak absorption at 1638 cm^−1^ and the height of the C=O vibration at 1710 cm^−1^ were determined. The degree of conversion was determined in accordance with the following equation:(1)$$\mathrm{Degree}\;\mathrm{of}\;\mathrm{conversion}\left(\%\right)=100\left[1-\frac{\left(\frac{A_{1638}}{A_{1710}}\right)\mathrm{polymer}}{\left(\frac{A_{1638}}{A_{1710}}\right)\mathrm{monomer}}\right]$$
where A_1638_ is the maximum height of the absorption peak at 1638 cm^−1^, A_1710_ is the maximum height of the absorption peak at 1710 cm^−1^, “polymer” refers to the cured specimen, and “monomer” refers to the uncured material.

### 2.3. Flexural Properties

The flexural strength of the materials was evaluated in accordance with the specifications provided by the ISO-4049 International Standard, while the elastic modulus was evaluated using the square section of the flexural mechanical properties [15,16]. Sixty bar-shaped specimens (25 mm × 2 mm × 2 mm) were prepared by placing the uncured samples into a stainless steel mold that was placed on a glass slide that was covered by a Mylar^®^ strip. A second strip and a glass slide were used to cover the mold. The samples were irradiated on both sides by the overlapping technique using the curing protocol previously described. After the polymerization process was performed, irregularities were removed using abrasive paper, and specimen dimensions were measured using a digital caliper (Mod. CD-6”C Mitutoyo. Tokyo, Japan). After storage in distilled water at 37 °C for 24 h, the specimens were placed in a universal mechanical test machine (Instron 4465, Norwood, MA, USA). A three-point flexural test was performed with a 1 kN load cell at a crosshead speed of 1.00 mm/min until a fracture occurred. The flexural strength (FS) and elastic modulus (EM) were calculated (in MPa and GPa, respectively) using the following equations:(2)$$\mathrm{FS}=\frac{3\mathrm{Fl}}{2{\mathrm{bh}}^{2}};\;\mathrm{EM}=\frac{F_{1}l^{3}}{4{\mathrm{bh}}^{3}d}$$
where F_1_ represents the load (N) exerted on the specimen on the linear portion of the load–curve deflection curve, F is the maximum load (N) exerted on the specimen at the point fracture, l is the distance (mm) between the supports, h is the height (mm) of the specimen, b is the width (mm) of the specimen, and d is the deflection corresponding to the load F_1_.

### 2.4. Water Sorption and Solubility

The hydric behavior of the materials was evaluated according to the specifications that are described in the ISO International Standard No. 4049. Sixty cylindrical discs (15 mm in diameter and 1 mm in thickness) were prepared in a stainless steel mold and polymerized following the curing protocol previously described. The samples were transferred to a desiccator, and their mass was monitored until a constant mass was obtained (m_1_; the loss in mass of each specimen was not more than 0.1 mg in a 24-h period). Then, the volume (V) of each specimen was calculated by measuring the thickness and diameter of the samples. The samples were immersed in distilled water at 37 °C for 7 days, and, after that, they were weighted to obtain m_2_. Finally, the samples were again placed in a desiccator and were weighted until their mass was found to remain constant (m_3_). The water sorption (WSP) and solubility (WSL) were calculated using the following equations:(3)$$W_{\mathrm{sp}}=\frac{m_{2}-m_{3}}{V};{\;W}_{\mathrm{sl}}=\frac{m_{1}-m_{3}}{V}.$$


### 2.5. Polymerization Shrinkage

The polymerization shrinkage (PS) was calculated using a linear transducer [17,18]. Disk-shaped specimens were prepared by placing the uncured composite (0.17 ± 0.04 g) at the center of a square cross-section brass ring (internal diameter 16 mm, height 1.24 mm), which was adhesively bonded to a glass microscope slide. The disk specimen was then covered with a glass microscope cover-slip. Then, the armature of a displacement transducer was carefully positioned to be in contact with the center of the cover-slip. Samples were photopolymerized according to the protocol described in Section 2.1. The cover-slip deflects when shrinkage occurs, so the deflection at the center of the cover-slip was monitored over time by the transducer, which has a sensitivity that is better than 0.1 mm. The transducer was connected to a signal and data acquisition unit. The shrinkage–strain deflection is defined by the following equation:(4)$$\mathrm{PS}=\frac{100\left(L_{0}-L_{t}\right)}{L_{0}}$$
where L_0_ is the height of the unpolymerized specimen (1.24 mm), and L_t_ is thickness of the polymerized specimen at the end of the polymerization process.

### 2.6. Statistical Analysis

The statistical analysis was performed using IBM SPSS Statistics 20 Software (Armonk, NY, USA). The data were evaluated to check for a normal distribution and homogeneity in variance. Analysis of variance was used to evaluate the effect of the experimental variables (radiant exposure and material) on the degree of conversion, flexural strength, elastic modulus, water sorption, water solubility, and polymerization shrinkage. The level of significance was set at p < 0.05.

## 3. Results

Figure 1, Figure 2 and Figure 3 show the flexural properties, polymerization shrinkage, water sorption, solubility, and degree of conversion for the dental resin composites and the three different light exposure protocols.

To analyze the effect of the radiant exposure and the applicability of the exposure reciprocity law, an independent one-way ANOVA analysis was performed after pooling the data for each composite. For the Admira^®^ composite, statistically significant differences between the light exposure protocols were observed for the elastic modulus (p < 0.001), and a higher elastic modulus was achieved when a higher radiant exposure was used; this was also observed for the Kalore^TM^ composite. For the Filtek^TM^ Z350XT composite, differences were only observed for the degree of conversion variable (p < 0.001). In this case, a higher radiant exposure allowed us to obtain higher degree of conversion values. The Tetric^ ®^ N-Ceram Bulk Fill analysis showed statistically significant differences for the elastic modulus and the water sorption (p < 0.001); specimens that had been exposed to radiant exposures of 10 J/cm^2^ showed higher values.

The preliminary results showed that most of the materials are governed by the exposure reciprocity law. Considering that all composites that were analyzed in this study have identical clinical indications, a one-way ANOVA analysis of the pooled data for each composite was conducted (Table 2). According to this second analysis, the Filtek^TM^ Z350XT composite showed the highest flexural strength, elastic modulus, and water sorption values (p < 0.001), while the Kalore^TM^ composite showed the lowest solubility, degree of conversion, and shrinkage (p < 0.001).

## 4. Discussion

According to the results, the exposure reciprocity law was upheld in the composites for almost all of the evaluated properties. In almost all cases, no relationship was observed between the radiant exposure and the analyzed chemical and physico-mechanical properties. Therefore, the null hypothesis of this study can be partially accepted.

Our results are consistent with previous works in which the exposure reciprocity law was found to be upheld for some properties [8,14,19]. In the photopolymerization process, the irradiance and light-curing time are two important factors that have an impact on the number of photons that are delivered to a specimen [20]. It has been accepted that the reciprocity law holds for most photochemical processes at reasonable light intensities [21], while at higher irradiance levels this reciprocity does not exist [19]. In this study, two levels of irradiance (400 and 1000 mW/cm^2^) have been used, and these levels of light intensity were not found to have any significant effect on the evaluated properties. It seems that the combinations of different irradiance and time protocols that have been used were not enough to induce a decrease or an increase in the number of radical growth centers during the early stages of polymerization, which could lead to differences in the material properties [4]. Consequently, the similarity in flexural strength, water sorption, solubility, and polymerization shrinkage among different curing protocols can be explained by the fact that the same degree of conversion was achieved in all cases. The degree of conversion is considered to be a very important material feature, and it is strongly correlated to some other material characteristics, including flexural strength and elastic modulus [22], polymerization shrinkage [23], and water sorption and solubility [24].

Among all the evaluated properties, the elastic modulus increased when higher levels of radiant exposure were delivered to the composite materials. During the photopolymerization of the dental composites, the degree of conversion and the crosslinking density increase rapidly, resulting in a rapid increase of the system’s viscosity that reaches a change of state called gelation, in which the polymer matrix becomes rigid [25]. In this phase, the development of the elastic modulus is the basis for the formation of shrinkage strains and stresses because the polymer shrinkage is directly transferred to the tooth structure [26]. It has been accepted that the method by which light energy is delivered to the material is capable of delaying the gelation of a composite, and therefore several methods of light modulation have been proposed with the objective of minimizing the stress that is generated by the volumetric shrinkage. In this study, the group with a radiant exposure of 5 J/cm^2^ presented lower values of the elastic modulus, which indicates that lower rates of conversion would allow for a better flow of the materials before they transition into the so-called gel state, which can lead to decreased contraction stresses while maintaining the other mechanical properties. Also, it is worth mentioning that the materials’ acquisition of the gel phase at early stages could be detrimental to the materials’ properties, since, once the materials are in the gel phase, the diffusion of free radicals through the material could be affected.

Interestingly, the Tetric^®^ N-Ceram Bulk Fill in Group B had the lowest values of water sorption. Although the values could not be correlated with the degree of conversion, it should be emphasized that the lower values achieved demonstrated a more cross-linked polymeric network, and more resistance to hydrolytic degradation is expected [24]. Since bulk-fill-type materials possess more translucency than conventional resin-based composites, the diffusion of light through the material is higher, and, consequently, more efficiency from the polymerization material is expected [27]. This feature seems to play an important role in the stability of these materials and should be further explored.

As the different curing protocols were not found to have any significant statistical differences in most of the evaluated properties, a second analysis was performed in order to establish which composite had a better performance. The results of this analysis are depicted in Table 2. With regard to flexural properties, the Filtek^TM^ Z350 XT composite had the highest flexural property (flexural strength and elastic modulus) values when compared with the other composites. These results are in agreement with previous studies, in which conventional composites had a higher flexural strength and a higher flexural modulus than bulk-fill composites [28], low-shrink composites [29], and ormocer-based composites [30]. These properties are very important features to study in dental composites in order to ensure that these materials, especially when used as posterior restorations, are not subject to the action of chewing forces that might induce permanent deformation [31]. The elastic modulus is directly related to the volume fraction of filler [32], so composites with higher filler volumes, such as the Filtek^TM^ Z350 XT composite, are expected to be stronger than those with lower filler volumes. Also, the use of an organic matrix that is composed of monomers with stiffer backbones could probably help achieve greater flexural properties [33]. Kalore^TM^, Admira^®^, and Tetric^®^ N-Ceram Bulk Fill presented lower flexural strength values when compared with Filtek^TM^ Z350, which suggests that these restorative materials require further study in order to fully understand the effect of their composition on their mechanical performance.

The phenomena of water sorption and solubility are important properties to study, since they may serve as precursors to a variety of chemical and physical processes, such as swelling, plasticization, softening, and hydrolysis, which can compromise the long-term mechanical properties of the materials [24]. The hygroscopic and hydrolytic characteristics of dental composites mainly depend on many factors that are related to the polymeric network structure, including the hydrophilic characteristics of the monomers it is built from [34]. The monomer TEGDMA has oxygen atoms in its ether linkages that are strongly hydrophilic and this could be the reason that Tetric^®^ N-Ceram Bulk Fill achieved the lowest water sorption. This composite does not include the TEGDMA monomer in its composition, and therefore water is not attracted to the polymer matrix composite. On the other hand, the Filtek^TM^ Z350XT composite achieved the highest values of water sorption. This behavior could be explained by the presence of the poly(ethylene glycol) dimethacrylate monomer within its polymeric matrix (Table 1), which possesses hydrophilic functional groups. In spite of the diversity in the hygroscopic behavior of the materials, it is very important to mention that, regarding water sorption and solubility, all evaluated materials exhibited values below 40 µg/mm^3^ and 7.5 µg/mm^3^, respectively, thus satisfying the conditions that are established in Specification ISO 4049.

The polymerization shrinkage in composites is directly related to the degree of conversion [35]. The values obtained in this study are in accordance with the literature for studies in which the polymerization shrinkage is evaluated through methods where the linear polymerization is calculated [28,36,37]. The Kalore^TM^ material exhibited the lowest polymerization shrinkage and degree of conversion values. The organic matrix of this composite contains a mixture of urethane dimethacrylate, dimethacrylate co-monomers, and the DX-511 monomer. The DX-511 monomer has a molecular weight of 895 g/mol, which is twice the molecular weight of Bis-GMA or UDMA. It has been shown that the magnitude of volumetric shrinkage is mainly determined by the number of covalent bonds that are formed as well as by the size of these molecules [17]. The larger the molecules for a given material volume, the smaller the number of double bonds, and, thus, the smaller the polymerization shrinkage [38]. A reduction in the polymerization shrinkage values in Kalore^TM^ is a desirable property, and would ensure that there is less polymerization stress, and a lower number of marginal defects and fractures, within the composite [39]. Nevertheless, lower values of double-bond conversion involve a residual monomer being trapped in the composite, which may reduce its biocompatibility and clinical serviceability [40].

## 5. Conclusions

Within the confines of this study, it can be concluded that, in most cases, the exposure reciprocity law was upheld at values of 10 J/cm^2^ of radiant exposure. The different chemical and physico-mechanical properties that have been analyzed were not affected by the different radiant exposure values used in this study. It seems that other RBC characteristics, such as the organic matrix’s composition, the type of photoinitiator, and the size and filler volume, have more of an influence on the materials’ properties.

## Figures and Tables

**Figure 1 dentistry-06-00055-f001:**
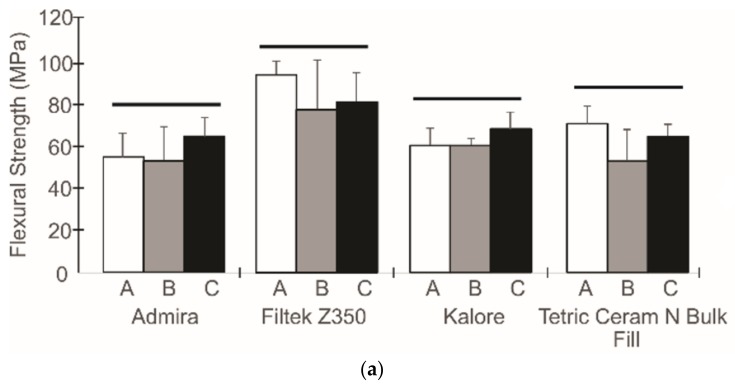
The flexural strength (**a**) and the elastic modulus (**b**) from different curing protocols. The columns under the same horizontal line are not statistically different.

**Figure 2 dentistry-06-00055-f002:**
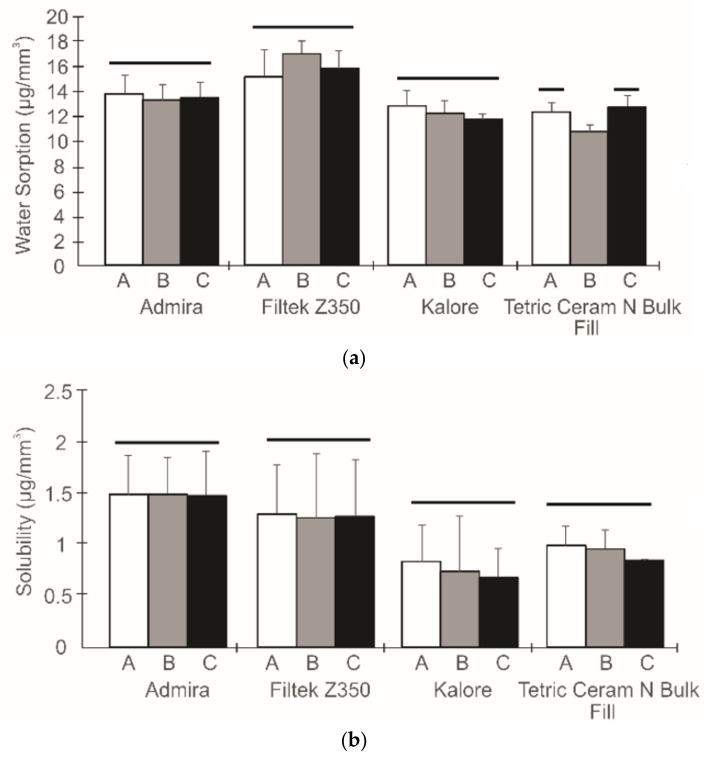
The water sorption (**a**) and the solubility (**b**) from different curing protocols. The columns under the same horizontal line are not statistically different.

**Figure 3 dentistry-06-00055-f003:**
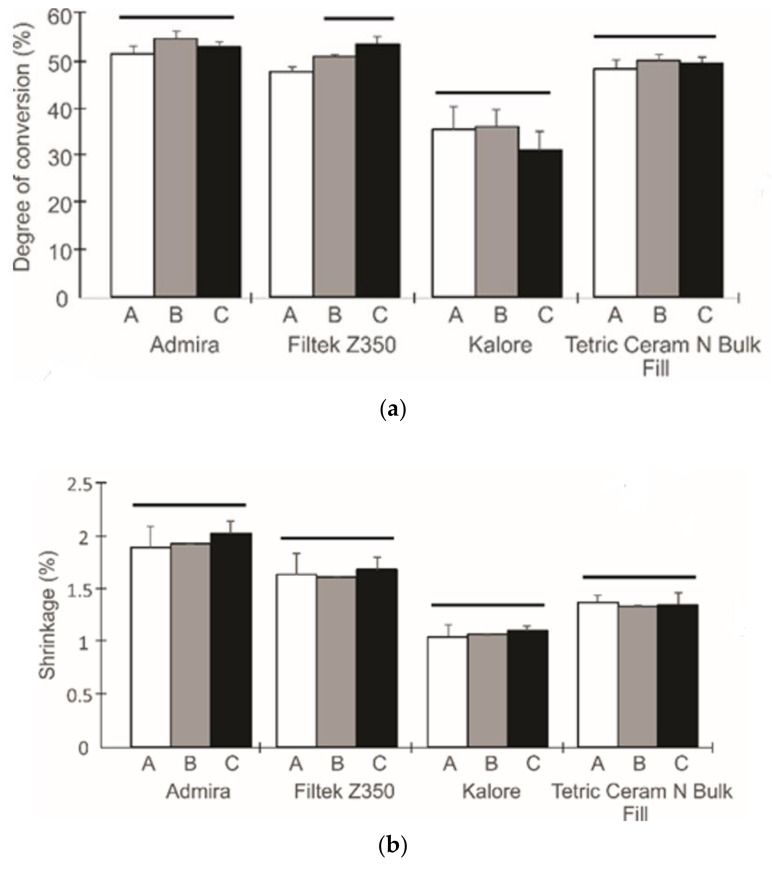
The degree of conversion (**a**) and the polymerization shrinkage (**b**) from different curing protocols. The columns under the same horizontal line are not statistically different.

**Table 1 dentistry-06-00055-t001:** Information on the manufacture of the resin-based composites.

Material	Organic Matrix *	Inorganic Filler *	Filler Load %Weight (Volume)
Kalore^TM^ (GC Corp., Tokyo, Japan)	UDMA, Urethane DX-511, dimethacrylate comonomers	Fluoro-aluminium-silicate glass, Prepolymerized filler, and Silicon dioxide	82 (55)
Admira^®^ (Voco, Cuxhaven, Germany)	Polisiloxane, aliphatic and aromatic monomers	Apatite-sulphate-phosphate and inorganic glass particles	78 (56)
Tetric^®^ N-Ceram Bulk Fill (Ivoclar-Vivadent, Schaan, Liechtenstein)	Bis-GMA, DMA	Barium-aluminium-silicate glass, ytterbium fluoride, and spherical mixed oxide	79 (60)
Filtek^TM^ Z350 XT (3M ESPE, St. Paul, MN, USA)	Bis-GMA, UDMA, TEGDMA, Bis-EMA, PEGDMA.	Silica, zirconia, and aggregated zirconia/silica clusters	72.5 (63.3)

* Information provided by the manufacturer. UDMA, urethane dimethacrylate; Bis-EMA, bisphenol A-glycol dimethacrylate; Bis-GMA, bisphenol A-glycidyl dimethacrylate; TEGDMA, triethylene glycol dimethacrylate; Bis-EMA, ethoxylated bisphenol a dimethacrylate; PEGDMA, Poly(ethylene glycol) dimethacrylate.

**Table 2 dentistry-06-00055-t002:** The pooled properties of the evaluated resin-based composites.

Material	Flexural Strength (MPa)	Elastic Modulus (GPa)	Water Sorption (µ/mm^3^)	Solubility (µ/mm^3^)	Degree of Conversion (%)	Shrinkage (%)
Filtek^TM^ Z350XT	80.52 (15.88)a	9.13 (0.66)a	15.93 (1.70)a	1.00 (0.60)ab	50.96 (2.77)ab	1.66 (0.15)b
Kalore^TM^	59.63 (10.13)b	7.85 (0.89)b	12.30 (1.01)bc	0.74 (0.36)b	33.72 (4.18)c	1.07 (0.08)d
Tetric^®^ N-Ceram Bulk Fill	60.37 (11.05)b	7.05 (0.60)b	11.85 (1.08)c	0.96 (0.26)ab	49.50 (1.53)b	1.36 (0.08)c
Admira^®^	54.17 (14.20)b	7.76 (1.11)b	13.49 (1.22)b	1.39 (0.47)a	53.36 (1.79)a	1.96 (0.15)a

The common corresponding letters (a–c) in a given column indicate that there are no significant differences.

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
