# Peer review of "Examining the Effect of Radiant Exposure on Commercial Photopolimerizable Dental Resin Composites"

_dentistry, 2018, doi:10.3390/dj6040055_

Round 1

Reviewer 1 Report

This manuscript reports the effect of light irradiation to the commercial composite resins, and authors conclude that mechanical properties were not altered at the high-dose irradiation investigated in this study. The topic of this paper should be of interest to the field of dental materials science. While this manuscript has potential interests to this field, there are some issues with the manuscript as submitted.

1) This manuscript is hard to read, because abbreviated and official terms are weaved out of context. In addition, so many abbreviated terms (e.g. FS, EM, WSP, WSK, PS) make manuscript more complicated. Furthermore, abbreviated term “DC” is given two interpretations (degree of conversion, and double bond conversion). The authors should double-check and revise throughout the manuscript.

2) There are too many errors in this manuscript, and this also make the manuscript unreadable. For example, numbered heading for Materials and Methods has mistakes, and numerical notation for L0 (2.4. polymerization shrinkage) should be wrong. In the last paragraph of Discussion section, what do the numbers “14” and “19” mean? I can’t give a full correction in this comment.

3) The authors present the result that Kalore RBCs indicate the highest elastic module in Group B. In contrast, Admira RBCs shows significantly lower elasticity in Group B compared with the ones of Group C. Tetric Ceram Bulk Fill RBCs show the significantly lower water sorption in Group B. But, Filtek Z350 RBCs have slightly higher water sorption in Group B compared with the other groups. The authors should give an adequate explanation for the difference in response to the same irradiation energy. Meanwhile, the very short SD bars in Group B for shrinkage data are unnatural and might be wrong.

4) In Discussion and Conclusion section, the authors insist that exposure reciprocity law is upheld for the RBCs in almost all properties evaluated in this study. I disagree with this opinion as far as looking at the data shown. This idea does make sense in general terms, but the authors may not conclude from your results, or otherwise should describe the substantial reasons based on the data shown in this study.

Author Response

Reviewer 1

This manuscript reports the effect of light irradiation to the commercial composite resins, and authors conclude that mechanical properties were not altered at the high-dose irradiation investigated in this study. The topic of this paper should be of interest to the field of dental materials science. While this manuscript has potential interests to this field, there are some issues with the manuscript as submitted.

1) This manuscript is hard to read, because abbreviated and official terms are weaved out of context. In addition, so many abbreviated terms (e.g. FS, EM, WSP, WSK, PS) make manuscript more complicated. Furthermore, abbreviated term “DC” is given two interpretations (degree of conversion, and double bond conversion). The authors should double-check and revise throughout the manuscript.

R. Thank for your comment. The abbreviated terms were eliminated, and instead, the full name of each property was evaluated. Also, the term degree of conversion was standardized along the text.

2) There are too many errors in this manuscript, and this also make the manuscript unreadable. For example, numbered heading for Materials and Methods has mistakes, and numerical notation for L0 (2.4. polymerization shrinkage) should be wrong. In the last paragraph of Discussion section, what do the numbers “14” and “19” mean? I can’t give a full correction in this comment.

R. Thank for your observation. The number heading for Materials and Methods was corrected. Also, the numerical notation for L0 was corrected. About numbers “14” and “19” in the last paragraph of the discussion section, they are references that were wrong formatted by the reference software manager used, these mistakes, were also corrected too.

3) The authors present the result that Kalore RBCs indicate the highest elastic module in Group B. In contrast, Admira RBCs shows significantly lower elasticity in Group B compared with the ones of Group C. Tetric Ceram Bulk Fill RBCs show the significantly lower water sorption in Group B. But, Filtek Z350 RBCs have slightly higher water sorption in Group B compared with the other groups. The authors should give an adequate explanation for the difference in response to the same irradiation energy. Meanwhile, the very short SD bars in Group B for shrinkage data are unnatural and might be wrong.

R. Thank you for your observation. The discussion section of the manuscript was improved, information about differences in response to the same irradiation energy was added.

4) In Discussion and Conclusion section, the authors insist that exposure reciprocity law is upheld for the RBCs in almost all properties evaluated in this study. I disagree with this opinion as far as looking at the data shown. This idea does make sense in general terms, but the authors may not conclude from your results, or otherwise should describe the substantial reasons based on the data shown in this study.

R. We decided to conclude that exposure reciprocity law is upheld for the RBCs in almost all properties evaluated due the lack of statistically significant differences observed among the groups, especially between groups B and C, were the same radiant exposure was delivered. To support this, we added more explanations in the discussion section. Thank you for your observation.

Reviewer 2 Report

This paper deals with current materials and important issues for clinicians.

There are several issues that need to be addressed.

1. The authors opted to test conditions other than those prescribed by manufacturers. This should be explained. Also, Group C is unlikely to happen in modern clinical practice.

2. The thickness of the samples, for all experiments, is rather low. These materials are expected to polymerise up to 4 mm and they are not tested to their limits.

3. The sorption measurements are very short in duration. A minimum of 1 month is expected and in some cases up to 6 months.

4. Depth of cure would have been a very relevant property to be tested here.

5. The authors must interact more extensively with current literature. There are two recent papers Ilie et al., Dent Mater, 2017, 880-94 and Ferracane et al., Dent Mater, 2017, 1171-91 that describe the requirements for particular testing for resin composites.

Most of these materials have been extensively characterized and the values quoted in this paper seem to be different than others in literature. This must be discussed. Flexural strength and shrinkage are rather low (see Shibasaki et al., Oper Dent, 2017, 42:E177-87).

Also, Alsunbul et al., Dent Mater. 2016 32(8):998-1006

Alshali et al., J Dent. 2015;43(12):1511-8

to name a few indicative references.

Author Response

Reviewer 2

This paper deals with current materials and important issues for clinicians.

There are several issues that need to be addressed.

1. The authors opted to test conditions other than those prescribed by manufacturers. This should be explained. Also, Group C is unlikely to happen in modern clinical practice.

R. Manufacturers’ recommendations about intensity used for photopolymerize their materials vary a lot. To cite an example, KaloreTM manufacturer (GC America) establish a minimum of 700 mW/cm2 during 20s, while for Admira, the value recommended is 300 mW/cm2. In this study, the values of 400 and 1000 mW/cm2 were chosen in function of the curing programs presets of the Bluephase® photopolymerization unit used. Unfortunately, these values are not user programmable and that’s the reason why we could not test the materials in the conditions, in terms of light intensity, prescribed by manufacturers.

Actually, we believe that the conditions tested in this study are valid, since most of the photopolymerization units currently available in the market does not allow to modify the light output of the LED source, and they are used by the clinicians in the intensities already established by the manufacturers.

In regards to group C, we decided to include a group using an intensity of 400mW/cm2 for several reasons. At first instance, this corresponds to the light intensity output of the Bluephase LED unit when actioned in the “LOW” mode. In addition, 400 mW/cm2 has been recommended as the minimum light intensity necessary to achieve adequate properties (please see DOI: 10.1016/j.dental.2014.12.010 and PMID: 8183730 references), which has confirmed in our study too.

2. The thickness of the samples, for all experiments, is rather low. These materials are expected to polymerise up to 4 mm and they are not tested to their limits.

R. Actually, we only tested one Bulk-fill composite (Tetric Ceram N Bulkfill, Ivoclar-Vivadent) and this represents the only material that can be polymerized in increments up to 4 mm. About the other materials tested, although the recommendations of each manufacturer vary, we noticed that none of them specify that the material can be used in the bulk-fill technique (increments up to 4 mm). Please find below the manufacturer’s instructions:

KaloreTM (http://www.gcamerica.com/products/operatory/KALORE/NEW_KALORE_10IFU.pdf)

Amira (https://www.voco.dental/es/portaldata/1/resources/products/instructions-for-use/e1/admira_ifu_e1.pdf)

FiltekTM Z350 XT (http://multimedia.3m.com/mws/media/631533O/filtek-z350-xtposterior-restoration-single-shade-technique-guide.pdf)

In addition, samples dimensions followed which is established in ISO International Standard No. 4049. To the best of our knowledge, evaluating flexural properties or water sorption and solubility altering the thickness of the specimens is not common. We believe that this feature could be explored in detail in future studies. Thank you for your comment!

3. The sorption measurements are very short in duration. A minimum of 1 month is expected and in some cases up to 6 months.

R. Besides it is not described in the text, the duration of the water sorption and solubility test was about 2 and a half months. Actually, the only period of time mentioned in the text is the storage period in distilled water to obtain the m2 value (seven days), which is specifeid in ISO International Standard No. 4049.

This issue related is probably due to a misunderstanding. However, we have rewritten the section 2.4 in order to facilitate the understanding of the methodology used. Please let me know if the modification makes clearer the methods used. Thank you!

4. Depth of cure would have been a very relevant property to be tested here.

R. We agree that the depth of cure is an important property that should be taken in count. However, as we evaluated only one bulk-fill material, which actually polymerizes in depth, we believe that the comparison with other materials, would be unfair.  This methodology will be considered for future studies. Thank you for your observation!

5. The authors must interact more extensively with current literature. There are two recent papers Ilie et al., Dent Mater, 2017, 880-94 and Ferracane et al., Dent Mater, 2017, 1171-91 that describe the requirements for particular testing for resin composites.

Most of these materials have been extensively characterized and the values quoted in this paper seem to be different than others in literature. This must be discussed. Flexural strength and shrinkage are rather low (see Shibasaki et al., Oper Dent, 2017, 42:E177-87).  Also, Alsunbul et al., Dent Mater. 2016 32(8):998-1006; Alshali et al., J Dent. 2015;43(12):1511-8; to name a few indicative references.

R. Actually, we had previously revised the manuscripts suggested (papers Ilie et al., Dent Mater, 2017, 880-94 and Ferracane et al., Dent Mater, 2017, 1171-91), and we agree that they bring valuable information for researchers who works with resin composites. In this case, such manuscripts suggest that the three-point bending test (ISO 4049) is the best method for evaluating both strength and elastic modulus. It is worth mentioning that these methodologies were used in the present study. About the polymerization shrinkage test used in this study, besides it is not indicated as the first choice, the use of linear transducer is considered as a valid method. References for this was added into the main text.

Also, the discussion section was modified and included information in an attempt to interact more extensively with current literature. Thank you for your observation.

Round 2

Reviewer 1 Report

This manuscript was considerably revised from previous version, and the authors had addressed the concerns from the original submission and improved its overall quality.

Author Response

We were pleased to know that this version of  our manuscript has adressed the concerns previously exposed. 

We would like to take this opportunity to express our sincere thank you for identifying areas of our manuscript that needed corrections. Your suggestions certainly contributed to improve the overall quality of the manuscript.

Reviewer 2 Report

The authors have addressed most of the comments.

In their response, they refer to some papers that they find useful but still do not cite in the revised manuscript.

Regarding the discussion and why flexural strangth and shrinkage values are low, still they do not offer an explanation. This is not a criticism for the values obtained or suggesting that the methodology used was not appropriate.

It has to do with discussing existing literature and highlighting the difference in values and suggest a possible reason why (probably methodological issues etc.)

Author Response

Reviewer 2

The authors have addressed most of the comments.

In their response, they refer to some papers that they find useful but still do not cite in the revised manuscript.

R. Thank you for the observation. The following references were now included in the manuscript:

Zorzin J, Maier E, Harre S, Fey T, Belli R, Lohbauer U, et al. Bulk-fill resin composites: Polymerization properties and extended light curing. Dent Mater. 2015;

Rueggeberg FA, Caughman WF, Curtis JW. Effect of light intensity and exposure duration on cure of resin composite. Oper Dent [Internet]. 1994 [cited 2018 Sep 29];19:26–32. Available from: http://www.ncbi.nlm.nih.gov/pubmed/8183730

Ilie N, Hilton TJ, Heintze SD, Hickel R, Watts DC, Silikas N, et al. Academy of Dental Materials guidance—Resin composites: Part I—Mechanical properties. Dent Mater. 2017;

Ferracane JL, Hilton TJ, Stansbury JW, Watts DC, Silikas N, Ilie N, et al. Academy of Dental Materials guidance—Resin composites: Part II—Technique sensitivity (handling, polymerization, dimensional changes). Dent Mater. 2017;

International Organization for Standardization. ISO 4049:2009 Dentistry Polymer based restorative materials. 2009.

Regarding the discussion and why flexural strength and shrinkage values are low, still they do not offer an explanation. This is not a criticism for the values obtained or suggesting that the methodology used was not appropriate.

It has to do with discussing existing literature and highlighting the difference in values and suggest a possible reason why (probably methodological issues etc.)

This paper deals with current materials and important issues for clinicians.

R. Thank you for the observation. For polymerization shrinkage, the method used in our manuscript (linometer), which measures linear polymerization shrinkage, shows relatively low values.

Our results are in agreement which those of some manuscripts where the polymerization shrinkage is evaluated through methods where the linear polymerization is calculated. The references for such manuscripts were added into the manuscript:

Han SH, Sadr A, Tagami J, Park SH. Internal adaptation of resin composites at two configurations: Influence of polymerization shrinkage and stress. Dent Mater. 2016;

Jang J-H, Park S-H, Hwang I-N. Polymerization Shrinkage and Depth of Cure of Bulk-Fill Resin Composites and Highly Filled Flowable Resin. Oper Dent. 2015;

Jung J, Park S. Comparison of Polymerization Shrinkage, Physical Properties, and Marginal Adaptation of Flowable and Restorative Bulk Fill Resin-Based Composites. Oper Dent. 2017;

With regards to flexural strength, explain the lower values obtained is beyond of our limits, since the methodology used to calculate these values strictly followed the specifications provided by International Standard ISO 4049; also, the intensity of the LED photopolymerization unit was always monitored using a radiometer (Bluephase meter, AG FL9494, Ivoclar-Vivadent).

Instead of that, we added some information where the same tendency of ours results was obtained, what it means that conventional nanocomposite had better performance than other type of composites (bulk fill, low-shrink and ormocer based). We believe that this information should bring relevant information to clinicians.